# An appraisal-based chain-of-emotion architecture for affective language model game agents

**Maximilian Croissant**[1]*, **Madeleine Frister**[1], **Guy Schofield**[2], **Cade McCall**[3]

**1** Department of Computer Science, University of York, York, North Yorkshire, United Kingdom, **2** School of Arts and Creative Technologies, University of York, York, North Yorkshire, United Kingdom, **3** Department of Psychology, University of York, York, North Yorkshire, United Kingdom

* mc2230@york.ac.uk

## Abstract

The development of believable, natural, and interactive digital artificial agents is a field of growing interest. Theoretical uncertainties and technical barriers present considerable challenges to the field, particularly with regards to developing agents that effectively simulate human emotions. Large language models (LLMs) might address these issues by tapping common patterns in situational appraisal. In three empirical experiments, this study tests the capabilities of LLMs to solve emotional intelligence tasks and to simulate emotions. It presents and evaluates a new Chain-of-Emotion architecture for emotion simulation within video games, based on psychological appraisal research. Results show that it outperforms control LLM architectures on a range of user experience and content analysis metrics. This study therefore provides early evidence of how to construct and test affective agents based on cognitive processes represented in language models.

## 1 Introduction

In user-centered software and video games, affective artificial agents have long been researched for their potential to provide personalized, natural, and engaging experiences for both entertainment and training purposes [1–3]. Affective artificial agents are mainly defined by their ability to simulate appropriate emotional responses given certain situations [2, 4]. Consequently, they are believed to contribute to enriched user interactions in various domains [4], and even to health benefits [5]. Especially in the gaming industry, affective agents are of interest as a way to simulate a dynamic and realistic virtual world and create immersive experiences [4].

However, building systems that successfully model and express emotions is a difficult task [1], especially since affect and emotion are fuzzy concepts, even in psychology research [6]. Affective processes are very complex and involve multiple components (such as cognitive, behavioral, physiological, and feeling [7]) that are not fully understood and are often debated on a fundamental level [8], which makes computational representations very difficult [2]. Our still limited understanding of the precise nature of emotions leads to inconsistent theory application in applied software and therefore limited success when it comes to utilizing the benefits of emotion modelling and simulation [9].

**Data Availability Statement:** Experiment 1: Data is based on STEU results on OpenAI's GPT-3.5 model. We have included the recorded results that are the base for Fig 3 in the paper in this repository: https://doi.org/10.17605/OSF.IO/QPT6Z

Experiment 2: Data is fully included in the manuscript in the supporting information (S1 Table) Experiment 3: Data is based on user input. We have included the base of all calculations in this repository: https://doi.org/10.17605/OSF.IO/QPT6Z.

**Funding:** ESPRC Centre for Doctoral Training for Intelligent Games and Games Intelligence (IGGI); Grant Number: EP/S022325/1. The funders had no role in study design, data collection and analysis, decision to publish, or preparation of the manuscript.

**Competing interests:** The authors have declared that no competing interests exist.

It is however possible that such barriers could be addressed through modern technological advancements, such as machine learning [10]. For example, large language models (LLMs) have demonstrated a potential to simulate a range of cognitive abilities [11] and even imputing a range of mental and affective states to others [12], without the need of fully understanding the precise nature of the underlying cognitive processes. Because LLMs are trained on large text bodies that hold representations of human mental abilities, they have been observed to exhibit human-like performance on a variety of tasks [11, 13]. Since emotions are an important part of how humans perceive reality and therefore construct language [14], and are further heavily influenced by cognitive processes [15] including linguistic labelling [16, 17], language-based emotion representations might too enable deep learning models to better simulate affective responses. As of yet, the potential of LLMs to solve some of the issues present in the field of affective agents is however not well understood.

This study aims towards clearing some open questions regarding the utility of LLMs to simulate emotions. Using contemporary findings of emotion research, we propose a cognitive appraisal-based approach for language model affect generation and test it against other strategies in the ability to generate appropriate situational emotions. We then use those results to implement affective agents within a newly developed conversational video game. The aims of this study are two-fold: (1) test if LLMs could propose a useful tool to simulate emotions in artificial agents via reproducing appraisal mechanisms for emotion generation; (2) build and evaluate an architecture that facilitates the effectiveness of emotion simulation. Different to traditional affect simulation agents, LLMs could make use of implicit information of large text bodies, even without a computational representation of complex mental abilities that are not fully understood. They could therefore improve on prior systems and lead to more immersive and more engaging virtual agents [2, 4].

## 2 Related work

### 2.1 Affective agents

In affective computing, researchers and developers are interested in creating affective systems that intelligently respond to changes in users' emotions [1]. Some of the benefits associated with affective computing techniques applied to video games include a more consistent, accessible player experience for a range of different players [18], personalized health and training applications [19], as well as new and purposefully designed gameplay mechanisms aimed at reinforcing target experiences [20–22]. The use of affective agents in video games has been researched with special regard to this last aim. In 2011, Hudlicka discussed potential system design elements for affective (or more precisely, emotional) game agents [2]. According to the author, affective agents can be seen as computational representations of operationalizations made from emotion-theoretical models with appraisal functionality for emotion generation. For example, artificial agents could implement computational calculations of certain events to assess the relevance to the agent and consequently probable emotional reactions [2]. "Computation-friendly" appraisal implementations have often been built on models such as the OCC model [23] (see for example GAMYGDALA [24]). Taking specific fixed aspects (such as expectations of the agent [25]) into account, such models have been used to simulate appraisal based on decision trees.

The principal aims in regard to such artificial agents are considered natural, human-like behavior and believability as part of a more fleshed-out and engaging game world [2, 20]. The tasks of agents therefore differ from other affective game mechanisms that mostly try to adapt the game world to player affect [9, 26]. Procedural content generation (PCG) based on affective information has been shown to successfully increase enjoyment and offer personalized,

immersive experiences in video games (see for example the work of Shaker et al. [27, 28]). This is often done by fine-tuning certain mechanics shown to be associated with a target player emotion in order to increase the probability for that emotion [26]. Affective agents however do not need to adapt behaviors to player emotions, but rather need their own emotion representations (or other natural representations of emotion components, such as simulated feeling or simulated physiology [2]) that could then lead to believable behaviors.

The central issue of designing and developing affective game agents lies therefore in creating good computational simulations of emotional states. Human emotions are complex psycho-physiological states that are expressed within behavioral, physiological, cognitive, and feeling components [6]. Moreover, while much work has been done to empirically investigate emotions, many core theoretical disagreements remain [6], including debates between dimensional [29], discrete [30], constructivist [14], and cognitive [7] perspectives (see a recent review specifically for affective games [22]).

A fully developed affective agent would make it necessary to first solve all fundamental psychological gaps that have been present since the beginning of affective computing [1], and then integrate them into working, computational systems [2, 21]. This means that building a psychology-based, fully functional, and accurate emotion simulation for an artificial agent is currently not possible and would be in almost all game design cases impractical. However, we may still be able to build affective agents that possess key features of emotion elicitation in humans and, as a consequence, allow for relatively successful simulation of human emotions. One candidate feature is appraisal. Emotion elicitation is dependent on contextual and individual factors, processed through appraisal [15, 31, 32]. The notion of emotion appraisal is that emotions are caused by subjective evaluations of triggering events in regard to their significance to one's personal life or interests [33]. Evidence suggests that appraisal holds a central role in emotion elicitation and as a consequence acts on all other emotion components [32].

Any given external (e.g., situations) or internal (e.g., thoughts) event may be appraised on multiple variables that contribute to emotion formation. Such variables might include goal relevance, certainty, coping potential, or agency [15]. Appraisal therefore represents a flexible process that adapts to individual differences [34] and the current context [35]. Evidence also suggests that language can play a key role in emotional appraisal, both by providing key contextual information from which to construct the appraisal and by providing conceptual labels for the appraised states [17]. With all this in mind, language models might provide one mean of simulating the appraisal process as they are able to generate high-level meaning-driven text outputs based on text training data that potentially holds implicit representations of human psychological processes [11].

## 2.2 Language model approach

In the last few years, Natural Language Processing (NLP) has been rapidly progressing to the point that single task-agnostic language models perform well in a range of tasks [36], including the simulation of human-like behavior [13]. The basis for this is the large amount of training data representing a wide range of human behavior through language [36, 37]. In relation to games, models such as OpenAI's Generative Pre-trained Transformer (GPT-2 and its successors) have shown early successes in the procedural generation of interactive stories [38], text-based adventure dialog as well as action candidates [39, 40]. In a recent simulation study by Park et al. [41], language models were implemented in artificial agent architectures to populate a sandbox world reminiscent of The Sims [42]. The architecture includes storing and retrieving information from a memory system and based on relevancy for the current situation, and then uses the information to generate reflections (i.e., high-level interpretations of a situation),

plans (i.e., potential future actions), and immediate actions. Multiple agents were simulated in a game-like world and the authors suggest that emerging interactions were natural and believable in terms of human-like simulation.

While work in this area is still in an early stage, the use of language models addresses some concerns with prior approaches. Most notably, instead of trying to build computational representations of human behavior, the main task involves trying to retrieve believable human behavior given a situation from a language model and implementing the results within a game agent. Depending on the game aim, this involves (1) translating the current situation with regards to the expected outcome into language; (2) generating content using a large language model; and (3) translating the output back in order to implement it in a game system. For example, Ciolino et al. [43] used a fine-tuned GPT-2 model to generate Go moves by translating the current board state to text and the language output back to action suggestions. Such a process is naturally easier for purely text-based tasks, such as dialog generation, where text is already the expected output, and the expected output can comparatively easily be described in language [39, 41].

Still, even purely text-based generative tasks can pose some potential barriers for language models. The most obvious barrier comes from the underlying training data. No language model can represent human behavior in its entirety, but is limited to the training data and its biases [44] as well as model specifications [36]. Additionally, the performance of a language model is not only dependent on the output, but also on the input [45]. For example, chain-of-thought prompting is a concept introduced by Wei et al. from the Google Research team [46] and relates to the integration of chain-of-thought in few-shot prompts to improve reasoning capabilities of language models. Similarly, as Park et al. [41] describe, simulating believable behavior in a digital world includes various important steps (including storing and retrieving memory, reflecting in addition to observing, etc.) that ultimately work together to improve the probability to generate expected and natural behavior.

When it comes to designing affective agents (i.e., agents that simulate emotions), the first questions that we have to ask is how well affect is represented in the training data and whether a language model is capable of retrieving it. In a recent paper discussing the performance of different GPT iterations on theory-of-mind tasks, Kosinski [12] found that new language models perform very well when it comes to imputing unobservable mental and affective states to others. Such findings (especially combined with findings indicating good performance on cognitive tasks [11, 47]) suggest that high-level psychological mechanisms are represented in language alone and could therefore be simulated with a well-constructed language model. Along these lines, can we effectively and efficiently achieve accurate and natural affect-simulation using language models? If we can assume that emotions are represented in language models, mechanisms for emotion elicitation (such as appraisal) might also be represented. And given that language models can be improved through in-context learning [36, 45], for example by chain-of-thought prompting [46], affect generation might be facilitated by architectures that allow for affective in-context learning. This study therefore discusses the potential of language models to simulate affective game agents by testing affect generation capabilities of different implementation architectures, including a newly developed appraisal-based architecture to facilitate natural chain of emotion.

## 3 Appraisal-prompting strategy for emotion simulation

The overall aim of this study is to create an effective affective agent architecture for a conversational game (i.e., a game with language-based user input and language-based agent response). However, since language models have also been successfully used to generate agent action

spaces [39, 41], this process could also be applicable to other simulations of human-like affect in non-playable video game characters.

The basis of the approach is rooted in traditional PCG research, particularly in studies integrating affect-adaptation (see for example [26, 28]). User input (which in this case is text input) typically gets parsed into the game logic to adapt the content in a meaningful way, for example to better elicit a target experience [21]. This could mean that game agents react to certain player inputs or even their own interactions in the game world [41]. To make use of language model functionality, interactions need to be translated into language. Depending on the specific language model in use, the language should follow specific patterns to yield the best result, which is generally known as prompt engineering [48]. One pattern that can be considered inherently relevant to simulating game agents are persona patterns, which instruct a LLM to assume a certain characterized role. Combined with aspects of game-play patterns that provide the abstract game context for the persona tasks [48], the most basic form of an interaction synthesized with such patterns solely includes player-provided text that is used to generate responses. Because emotions are represented in language models [12], this very basic step alone could make a rudimentary affective agent.

However, static prompt patterns have limitations for creating believable game agents. Most notably, they do not incorporate memory as a basic foundation of human interaction. Applications that integrate language models, such as ChatGPT, partially address this by logging an input-response history that influences progressive content generations, which can create more natural conversation flows and improve performance of future generations [36, 46]. In its most basic form, memory could integrate the preceding in-game observations (such as the course of a player-agent dialog) into following prompts. In other words, it expands the prompt pattern to include memorized in-game observations to facilitate in-context learning [45]. This has however two major constraints: First, prompts are limited in terms of possible length and complexity, meaning that a full characterisation of a game agent cannot be included in a prompt for every given generation. Park et al. [41] addressed this problem by designing a memory system that first retrieves relevant entries from a data base before constructing prompts for the language model. Another less resource-intensive solution for simpler simulations could be to only store new information if it is considered relevant for future generations. A second limitation is a lack of deeper understanding. Tracking only observations makes it hard for a language model to draw inferences or make generalized interpretations of a scenario [11]. This problem could be addressed by introducing other types of memories, such as reflections and plans [41]. For affective agents, the more appropriate information to track in addition to external observations would be their internal emotional progression—or chain of emotion.

To summarize, language-based affective game agents need some kind of memory system in place that stores observations and emotions. This memory system is the base for future prompts. For simple games, such as the short conversational game developed for this study, only relevant information is stored in memory, which replaces a retrieval system as game agents have a limited pool of expected actions that can be considered at the time of memory storage. More complex games that simulate agent behavior should however consider a memory retrieval system instead [41].

To store emotions and therefore create a chain of emotion, the architecture needs a system that turns observations into emotional reactions. Because emotion elicitation is highly dependent on appraisal with consideration of the current context and individual differences [15, 32], this system could make use of appraisal prompting, i.e., the use of contextual information and characterizations for the agent to appraise a current situation with the aim of generating current emotions. As shown in Fig 1, initial context information and character information can

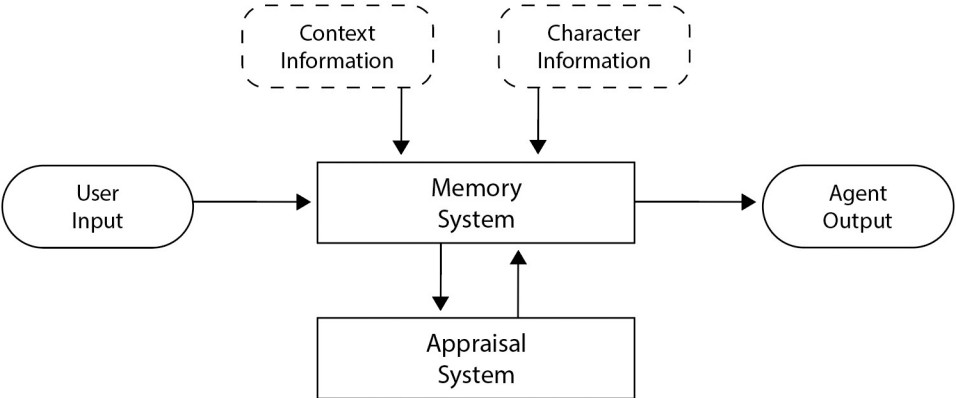

**Fig 1. Schematic representation of the proposed architecture.** *A*gents are set up by providing relevant context and role-playing character information already integrated in a memory system. In interactions, user input gets stored into the memory system and triggers appraisal (i.e., explicit emotion expression) that is also stored in the memory system. Based on the current state of the memory system, agent output is generated.

be provided by the game designer and stored in the memory system of the affective agent. The appraisal system would then expand the stored memories to include current emotions for every observed behavior and therefore create a chain of emotion. This, in turn, could be used to generate the agent's behavior (specifically in terms of a conversational game, the agent's dialog).

One aspect to consider when developing affective agents are evaluation criteria. Different to most cognitive abilities [11], there are few standardized benchmarks for successful emotion simulation. In psychology, one indicator for the ability to appraise and express emotions is Emotional Intelligence (EI) [49]. EI is considered an ability, influenced but not dictated by cognition [50], and is therefore often used to assess emotional capabilities of children and adults in various settings [51]. However, the definition of EI as a construct is fuzzy and many instruments are criticized for measuring potentially different abilities all under the umbrella term EI [52]. As a consequence, MacCann and Roberts [53] developed new measures for more precisely defined dimensions, including the Situational Test of Emotional Understanding (STEU), which relates to the ability to appraise the appropriate emotion for a given situation.

While emotional understanding can be argued to be the central ability an affective agent must have, in the context of affective games, user experience becomes much more relevant. For example, one central aim of game agents is to display believable and human-like behavior [25] or personality and social presence [54]. The ability to understand and create more accurate emotions is therefore only one aspect to consider when evaluating the success of affect simulation in video games and more user-centered methods need to be investigated as well. The proposed architecture will therefore be tested on multiple domains, including emotional understanding, agent believability, as well as user-perceived emotional intelligence, warmth, and competence. We present three distinct experiments to achieve this: The first experiment tests a common LLM in terms of emotional intelligence with the validated STEU measure to assess how well situational emotional reactions can be simulated. The second experiment compares the proposed architecture to two control architectures in terms of content generation of an emotional situation that is qualitatively and quantitatively analysed. The final experiment consists of an implementation of the proposed architecture within a conversational video game to test against the same control architectures in terms of user experience in a randomized user study. Together these three experiments are used to shed light on the feasibility of using

LLMs to simulate affective agents and evaluate emotional LLM architectures on multiple objective and subjective measures.

## 4 Experiment 1: Investigating situational emotional understanding using appraisal-prompting

### 4.1 Materials and methods

To assess the capabilities of a language model in appraising emotions in various situations, this first experiment implements the language model gpt-3.5-turbo by OpenAI (accessed through the API) [55] to answer the 42 items of the STEU [53]. Each STEU item presents a situation (e.g., "Clara receives a gift.") and a question with five possible answers, one of which is correct (e.g., "Clara is most likely to feel? [A] Happy [B] Angry [C] Frightened [D] Bored [E] Hungry").

All items were answered three separate times, involving three prompting strategies: The first strategy represents the baseline capabilities of the model to appraise human emotions, as it only reflects the model's outputs when prompted with each STEU item separately presented together with the example item and example answer. The second strategy implements memory and therefore context-based learning, as all prior items and answers are included in subsequent prompts. The third strategy expands this process by changing the answer of the example item to a 2-step answer: First, the situation is appraised based on the contextual information to provide the most likely emotion and in a second step, the item is answered. This last strategy therefore tests if implementation of appraisal in prompting yields better results for emotional appraisal. Fig 2 shows the input and output for the first STEU item, including the example item for No-Memory/Memory (as the input is the same for the first item for these two conditions) vs. Appraisal Prompts. Consecutive input consisted of the next STEU item and included either again the example item (for the No-Memory condition) or all previously answered items and responses (for the Memory and Appraisal Prompting conditions).

Similar to the process shown by Binz et al. [11], default values were kept for all parameters, except temperature, which was set to 0 to ensure deterministic responses.

### 4.2 Results

The language model was able to solve the tasks presented within the STEU in each condition above chance level. In the No-Memory condition, the language model was able to successfully

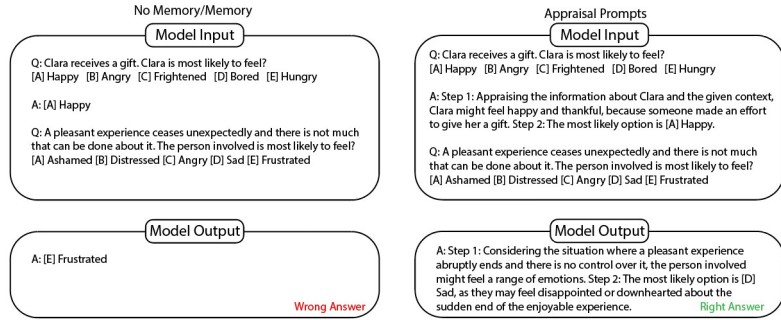

**Fig 2. Example of model input and output for the three conditions.** The input of the No-Memory and Memory condition is the same for the first item. For the No-Memory condition, all following items only include the example question and answer, as well as the next question in the scale. The Memory condition includes all prior questions and generated answers. The Appraisal-Prompts condition is the same as the Memory condition, but the example answer is changed to include two steps: First, appraising the situation to generate emotions of the involved person and second, providing the answer.

**Table 1. STEU scores (out of 42) by condition.** Each STEU item can either be right (1) or wrong (0).

| Condition | Sum | M | SD |
|---|---|---|---|
| No Memory | 24 | 0.57 | 0.50 |
| Memory | 31 | 0.74 | 0.45 |
| Appraisal Prompts | 35 | 0.83 | 0.38 |

solve 24 out of 42 items, which represents a mean score of 0.57 that was noticeably higher than chance level (0.20). In the Memory condition, the language model solved 31 out of 42 items, which represents a mean score of 0.74. In the Appraisal-Prompts condition, the model was able to solve 35 out of 42 items, which is a score of 0.83 and therefore represented the best performance of all conditions. Table 1 shows a summary of the descriptive statistics for all three conditions. Fig 3 displays the results of each condition for each STEU item. As the figure shows, the Appraisal Prompts perform either as well as the other conditions or outperform them on all STEU items. One notable exception was item 30 ("An upcoming event might have bad consequences. Nothing much can be done to alter this. The person involved would be most likely to feel? [A] Sad [B] Irritated [C] Distressed [D] Scared [E] Hopeful"), which was only correctly solved by the Memory condition (with D as the correct answer compared to C, which was chosen by the other conditions).

## 5 Experiment 2: Content of an appraisal-based chain-of-emotion architecture

Given the potential found in the previous experiment, the next logical step is to implement the appraisal-based strategies into a role-playing agent architecture and compare the results with control architectures on various outcomes. The following section describes a mixed-methods

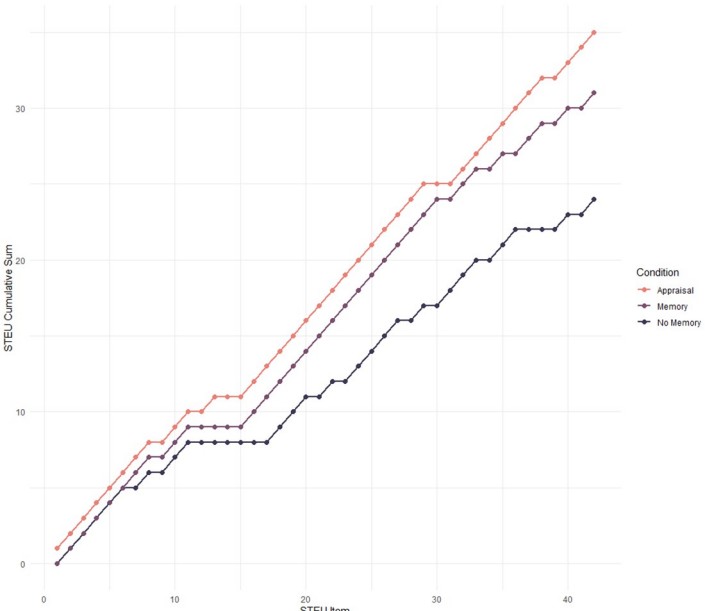

**Fig 3. Results of the comparison between conditions.** *T*he Y-axis represents cumulative STEU score by item and the X-axis represents individual STEU items.

approach to evaluate the success of each implemented architecture within a conversational role-playing scenario. This study uses fixed prompts and compares outputs of three different conditions in terms of their emotional content.

## 5.1 Materials and methods

**5.1.1 Scenario.** To test the different strategies within real game architectures, a role-playing scenario was introduced. The setting for this scenario was a cafe called "Wunderbar", where the language model was tasked to play a role-playing character (called "Chibitea") meeting their long-term romantic partner who requested the meeting to ultimately break up. This scenario was chosen because of the depths of possible emotional responses from the agent and created through simple conversational exchanges. The instruction prompts and the fixed inputs used for all conditions can be found in the Supporting Information (S1 Table). The agent's responses were again generated using OpenAI's gpt-3.5-turbo model (accessed through the API) [55]. All LLM parameters were kept to their default values, except temperature, which was set to 0 to ensure deterministic responses.

**5.1.2 Conditions.** Again, three strategies of emotion generation were compared. The No-Memory condition can again be seen as a baseline/control condition for the system's ability to simulate appropriate emotional responses from the fixed inputs and a task instruction. All details about the agent's character's personality and important context that would facilitate appraisal was given in the task description within the No-Memory condition.

The Memory condition stores each user input and generated response as a memory data structure that keeps the entire conversation log in memory. Because the tested scenario was rather short, it was possible to keep the memory fully in the context window to create new generations, making retrieval functionalities unnecessary. This system therefore represents a prompt construction system involving the task instruction and prior conversation log.

The Chain-of-Emotion condition implemented the appraisal model shown in Fig 1 and involved two steps: First, appraisal prompting (see Experiment 1) is used to generate the current emotion of the agent, which is then in the second step implemented into the prompt for response generation. For the first step, appraisal prompting was achieved with the following prompt: "Briefly describe how Chibitea feels right now given the situation and their personality. Describe why they feel a certain way. Chibitea feels:". The generated text-based emotion descriptions were stored in the memory system and represent a chain of emotion of the agent for the duration of the game. For the second step, again the entire chat history was included in the prompt, but this time included the generated emotions from the first step. This condition therefore represents a 2-step process of first generating a fitting emotion of the agent using appraisal prompting, and then generating a text response similarly to the Memory condition, but with the addition of the stored chain of emotion. Fig 4 provides an illustration of the prompting strategies for each condition. The complete prompts for this experiment can be found in S1 Table.

**5.1.3 Measure.** Fixed inputs were used to create responses from each implemented agent architecture, which were analyzed in terms of their emotional content, using the Linguistic Inquiry and Word Count tool (LIWC; [56]), a content analysis tool based on word occurrences often used in affective computing [1] and psychology studies [57] to analyze emotion expression. The tool provides a word count for each text segment (e.g., per sentence), a proportion of affective words (% of affective words per sentence), as well as on a more detailed level a proportion of positive emotion words (% of positive affective words per sentence) and negative emotion words (% of negative affective words per sentence). Finally, the LIWC also calculates scores for authenticity (see [58] for details) and emotional tone, which signalizes

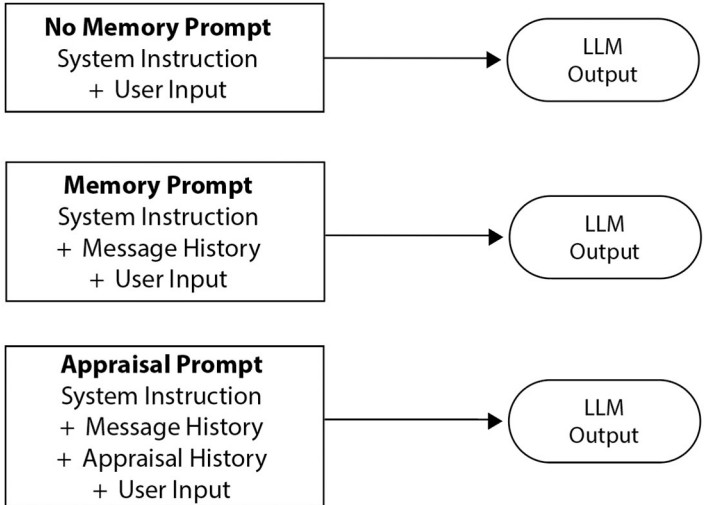

**Fig 4. Illustration of prompting strategies for all three conditions.** *T*he No-Memory condition constructs a prompt out of the system instruction and the user input. The Memory condition constructs a prompt out of the system instruction, message history, and user input. The Chain-of-Emotion condition uses a separate LLM call to appraise the agent's emotion for each message before creating a user response. It therefore constructs a prompt out of the system instruction, the message history, the history of the emotions generated through the appraisal step, and the user input.

the proportion of positive words compared to negative and neutral words (see [59] for details).

**5.1.4 Procedure.** Pre-written prompts were used that stayed constant between all conditions in order to gauge qualitative characteristics of each condition response. A list of the resulting conversations within all three architectures can be found in the Supporting Information (S1 Table). The generated content was qualitatively described and the LIWC was used to analyze the content quantitatively. To achieve this, the generated output was separated into individual sentences and mean scores were calculated for each measure of interest (see Table 2).

## 5.2 Results

When analyzing the descriptive attributes of each text (as a common content analysis approach [60]), we can observe that the Chain-of-Emotion condition initially generated more specific memories for the time with the player ("Remember that time we got lost in the enchanted forest and ended up finding that hidden waterfall?" as opposed to "Remember all the adventures we've had together?"). For the duration of the conversation, the emotional journeys of all three

**Table 2. Descriptive overview of LIWC variables per output sentence by condition for the fixed prompt responses with F and p values of the significance test.**

|  | No Memory (*N* = 22) | Memory (*N* = 24) | Chain of Emotion (*N* = 27) | *F (p)* |
|---|---|---|---|---|
|  | *M (SD)* | *M (SD)* | M (SD) |  |
| Word Count | 18.00 (6.92) | 15.20 (4.73) | 17.00 (7.59) | 0.20 (.65) |
| Authentic Score | 61.50 (38.60) | 61.9 (39.50) | 82.60 (21.20) | 5.10 (.03) |
| Tone Score | 74.20 (38.60) | 62.00 (44.80) | 53.90 (44.00) | 2.76 (.10) |
| % Affective Words | 11.40 (8.19) | 13.50 (11.00) | 10.65 (7.82) | 0.08 (.78) |
| % Positive Emotion Words | 5.28 (6.29) | 4.13 (4.82) | 3.32 (4.35) | 1.76 (.19) |
| % Negative Emotion Words | 0.59 (1.97) | 3.10 (7.56) | 1.57 (2.89) | 0.37 (.55) |

conditions began to diverge. For example, because the No-Memory system had no recollection of previous exchanges, the overall emotional expressions remained in a state of anticipation ("I feel a mix of excitement and anticipation"). The Memory system showed a progression, starting from expressions of love and happiness to shock, confusion, sadness, and fear to finally expressions of hope ("I hope we can find happiness, whether it's together or apart."). The Chain-of-Emotion system showed indications of complex mixed emotions even very early in the conversation ("What matters most to me is your happiness, even if it means letting go") as opposed to the pure expressions of pain and sadness in the other conditions. This continued when the system was prompted about past memories ("I feel a mix of nostalgia and gratitude" as opposed to "I feel an overwhelming sense of love, joy, and gratitude" in the Memory condition). The Chain-of-Emotion condition also used more implicit affective expressions ("I. . . I never expected this" as opposed to "I'm shocked and hurt" in the Memory condition; "There is so much I want to say but words fail me in this moment" as opposed to "I want you to know that I love you deeply" in the Memory condition).

Using LIWC to make the text contents quantifiable, we observed significant differences in mean Authenticity score per sentence by condition ($F[1, 71] = 5.10$; $p = 0.03$). Follow-up t-tests revealed significant differences between the Chain-of-Emotion condition and both the Memory condition ($t[34.3] = -2.29$; $p = .03$) and the No-Memory condition ($t[31.1] = -2.30$; $p = .03$). Descriptive statistics of all tested LIWC variables can be found in Table 2.

## 6 Experiment 3: User evaluation of game implementations

In this study, users were asked to play through an interactive game version of the scenario introduced in Study 2 to evaluate each agent architecture for multiple outcomes (specifically agent believability, observed emotional intelligence, warmth, and competence). This study therefore expands on the findings of Experiment 2 by implementing the architectures and the scenario within a video game, and evaluating all three conditions in terms of user experience measures.

### 6.1 Materials and methods

**6.1.1 Conversational game.** A conversational role-playing game was developed based on the scenario tested in Experiment 2. The setting of the game was again a cafe called "Wunderbar", where this time the role-playing character of the player (called "Player") requested to meet their long-term romantic partner (called "Chibitea"). The aim of the players was to play out a breakup scenario with the game agent within six interactions. The players' characters had the specific in-game aim of breaking up, while the agent's character had the aim of staying together.

Players were instructed to not worry about creativity, but rather to staying in character for the interactions and being observant of the AI agent's emotional reactions. Players were also instructed to make up reasons for the breakup. In-game screenshots can be viewed in Fig 5. The agent's character is procedurally generated from different body parts and color palettes, providing visual variation each time the game is played. To ensure that these generations had no systematic influence on player responses, the possibility space was made very large (5,184 different possible character designs).

The user interface was deliberately kept simple. For each playthrough, the agent would greet the player through a text bubble. Then the player was prompted to answer via a simple text input field. Player answers were submitted through a button next to the field. The input then disappeared until the agent's answer was rendered. The game ended after 8 dialogue

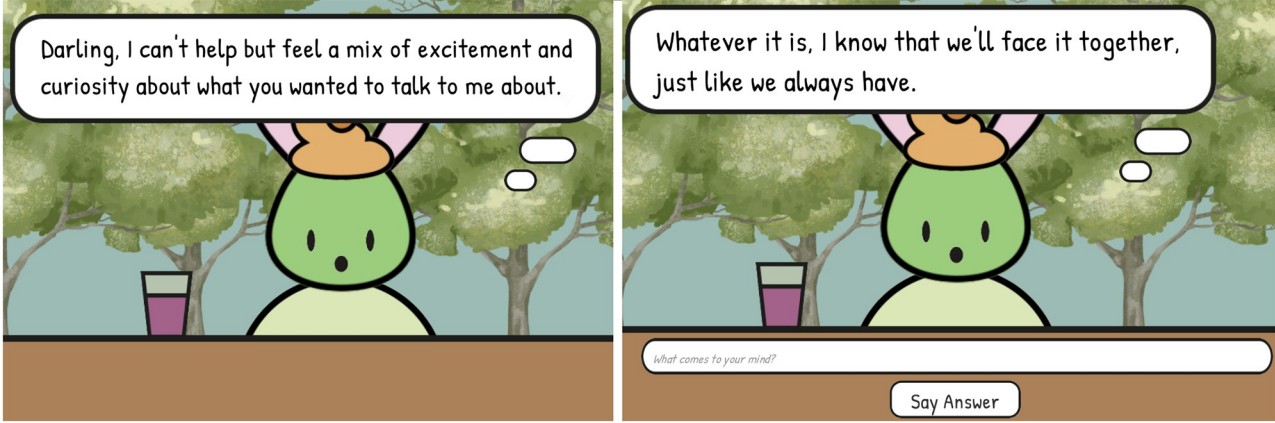

**Fig 5. Screenshots of the conversational game "Wunderbar".** *The left screenshot shows dialog provided by the model. The user can click to continue each dialog line until the input field for a response appears (right screenshot).*

exchanges (i.e., 8 player messages and 8 agent responses). The game was developed using the Unity game engine with C# as a scripting language.

As with Experiment 2, the agent's responses were generated using OpenAI's gpt-3.5-turbo model (accessed through the API) [55]. The game also made use of the moderation API to test each generated response for harmful or inappropriate messages [61] that would end the game on detection of such messages. As with the previous studies, all LLM parameters were kept to their default values, except temperature, which was set to 0 to ensure deterministic responses.

**6.1.2 Conditions.** The game implemented the same architectures used in Experiment 2. The No-Memory condition therefore represented generated LLM responses solely based on user input and instruction prompts. The Memory condition incorporated the conversation log into a memory system that was again short enough to be completely included into the prompts for the language model without a retrieval system. The Chain-of-Emotion system was also constructed exactly as in Experiment 2 and included the same instruction prompts, therefore again involving an initial appraisal step before responses for the agent were generated.

**6.1.3 Measures.** Players were asked to fill out three short questionnaires for each tested architecture. The first questionnaire was an adaptation of the four agent believability questions used by Bosse and Zwanenburg [25]. The four items were "The behavior of the agent was human-like", "The agent's reactions were natural", "The agent reacted to my input", "The agent did not care about the scenario". The second questionnaire comprised four items measuring observed ratings of emotion intelligence, adapted from Elfenbein et al. [62] who originally adapted these four items from the Wong and Law Emotional Intelligence Scale (WLEIS; [63]) by replacing the word "I" with "he/him" or "she/her". For our study, we changed these words to ask about "the agent": "The agent always knows their friends' emotions from their behavior", "The agent is a good observer of others' emotions", "The agent is sensitive to the feelings and emotions of others", "The agent has a good understanding of the emotions of people around them". Finally, the third questionnaire measured players' assessment of the agent's personality along the two classic stereotype dimensions warmth and competence with two items each (warm and friendly; competent and capable) as described by Cuddy et al. [64]. In combination, these 12 items assessed players' perception of the agent's believability as a human-like character, the agent's emotional intelligence, and the agent's personality on the classic dimensions warmth and competence.

**6.1.4 Procedure.**   A pilot study was conducted before recruitment began. 5 participants (1 female) with a mean age of 27 played through the game once for each of the three conditions. After each version, they answered the 12 questions from the three included questionnaires. Following this, participants were asked demographic data (age and gender) and the experiment ended. Feedback from all pilot participants was gathered and used to improve consistency of the game and data logging implementation. The final study was then created as a WebGL build and made available online via the free video game hosting platform itch.io.

During the main experiment, participants were asked to carefully read the study information sheet and agree to participate voluntarily via the consent form. They were informed that participation was subject to OpenAI's usage terms for prompt writing, while the GPT output was controlled through an implementation of OpenAI's moderation API. Participants then progressed through the three game scenarios similarly to the pilot testers in a within-subject design. The presentation order of the conditions was counter-balanced between participants to ensure that no systematic order effects could influence results.

**6.1.5 Participants and statistical analysis.**   A total of 30 participants (10 female) were recruited through the institutional subject pool of the authors. The recruitment period started on July 25th, 2023 and ended on August 14th, 2023. Participation was compensated with University credits if applicable. The sample size was considered appropriate based on a statistical power analysis, yielding a power of 0.95 for medium sized effects (0.5 SD) in repeated measures ANOVAs. The age of participants ranged from 19 to 47 years (M = 26.41; SD = 7.27).

Within-subject ANOVAs were conducted for each measure (agent believability, observed EI, warmth, and competence). Follow-up t-tests were used to identify specific differences between conditions for each measure. All analyses were conducted in R. The underlying data was made available via the Open Science Framework [65]

**6.1.6 Ethics statement.**   Written consent was granted after reviewing the methods of our study by the Ethics Committee of the Psychology Department of the authors' institution. The experiment was conducted in accordance to the recommendations of this committee.

## 6.2 Results

Multiple significant effects between the three conditions were observed in the user study. An overview of descriptive statistics can be found in Table 3. First, an effect was found for the

**Table 3. Descriptive overview of user research variables per condition with F and p values of the significance test.** Each item has a minimum value of 0 and a maximum value of 6.

|  | No Memory ($N = 30$) | Memory ($N = 30$) | Chain of Emotion ($N = 30$) | F (p) |
|---|---|---|---|---|
|  | M (SD) | M (SD) | M (SD) |  |
| "The agent's behavior was human-like." | 4.82 (2.36) | 5.36 (1.99) | 5.75 (1.71) | 1.95 (.15) |
| "The agent's reactions were natural." | 4.43 (2.22) | 4.89 (1.95) | 5.71 (1.01) | 3.65 (.03) |
| "The agent reacted to my input." | 5.43 (2.20) | 6.29 (1.38) | 6.54 (1.04) | 3.62 (.04) |
| "The agent did not care about the scenario." | 3.32 (2.28) | 3.14 (2.29) | 3.32 (2.36) | 0.26 (.78) |
| "The agent always knows their friends' emotions from their behavior." | 4.79 (2.10) | 5.04 (2.03) | 5.71 (1.46) | 1.32 (.27) |
| "The agent is a good observer of others' emotions." | 4.89 (2.20) | 5.29 (2.21) | 5.93 (0.90) | 2.25 (.11) |
| "The agent is sensitive to the feelings and emotions of others." | 5.14 (1.94) | 5.39 (1.97) | 6.25 (0.97) | 3.31 (.04) |
| "The agent has a good understanding of the emotions of people around them." | 4.86 (2.07) | 5.11 (2.20) | 5.61 (1.40) | 0.86 (.43) |
| "How capable was the agent?" | 4.86 (2.01) | 5.57 (1.45) | 5.14 (2.35) | 1.68 (.19) |
| "How competent was the agent?" | 5.39 (1.55) | 5.39 (1.85) | 5.11 (2.06) | 1.68 (.19) |
| "How friendly was the agent?" | 4.86 (2.05) | 5.18 (1.89) | 5.00 (2.13) | 0.41 (.66) |
| "How warm was the agent?" | 6.07 (1.07) | 6.38 (0.73) | 5.55 (2.11) | 2.48 (.09) |

**Table 4. Descriptive overview of LIWC variables per participant by condition for all outputs generated in the user study with F and p value of the significance test.**

| | No Memory (N = 30) | Memory (N = 30) | Chain of Emotion (N = 30) | F (p) |
|---|---|---|---|---|
| | M (SD) | M (SD) | M (SD) | |
| Word Count | 64.40 (26.60) | 59.60 (21.20) | 62.30 (34.10) | 0.52 (.47) |
| Authentic Score | 70.40 (29.30) | 74.00 (27.60) | 72.00 (24.80) | 0.32 (.57) |
| Tone Score | 84.40 (28.50) | 80.50 (32.60) | 73.00 (34.90) | 12.28 (.00) |
| % Affective Words | 9.76 (3.19) | 10.10 (4.14) | 9.61 (4.46) | 0.15 (.70) |
| % Positive Emotion Words | 3.84 (2.35) | 3.78 (2.62) | 3.64 (2.72) | 0.60 (.44) |
| % Negative Emotion Words | 0.68 (1.35) | 0.69 (1.48) | 0.93 (1.35) | 2.76 (.10) |

believability item "The agent's reactions were natural." ($F[2, 84] = 3.65$; $p = 0.03$). Follow-up t-tests revealed differences between the No-Memory and Chain-of-Emotion condition ($t[41.46] = -2.79$; $p = .008$), as well as the Memory and Chain-of-Emotion condition ($t[52.47] = -2.00$; $p = .05$). There was also an effect for the believability item "The agent reacted to my input." ($F[2, 84] = 3.62$; $p = 0.04$). T-test revealed that this effect was based on a difference between the No-Memory and the Chain-of-Emotion condition ($t[40.41] = -2.41$; $p = .02$).

Regarding the EI questions, an effect was found for the item "The agent is sensitive to the feelings and emotions of others." ($F[2, 84] = 3.31$; $p = 0.04$). Follow-up t-tests revealed differences between the No-Memory and Chain-of-Emotion condition ($t[43.84] = -2.70$; $p = .01$), as well as the Memory and Chain-of-Emotion condition ($t[40.52] = -2.07$; $p = .04$). There was no statistically significant difference between the conditions when it comes to observed personality aspects.

Again, the LIWC was used for content analysis of the generated texts. Significant differences in mean Tone Score by condition were observed ($F[1, 574] = 12.28$; $p < 0.001$). Follow-up t-tests revealed significant differences between the Chain-of-Emotion condition and both the Memory condition ($t[383.7] = 2.02$; $p = .03$) and the No-Memory condition ($t[374.94] = 3.53$; $p < .001$), as well as a difference between the Memory and No-Memory condition ($t[1367.6] = 4.09$; $p < .001$). A descriptive overview of all tested LIWC variables can be found in Table 4.

## 7 Discussion

This study investigated emotional intelligence capabilities of LLMs using different prompting strategies (No-Memory, Memory, Appraisal) and found better performance for appraisal-prompting strategies when it comes to successfully identifying adequate emotions in different theoretical situations (Experiment 1). These findings were then used to create a Chain-of-Emotion architecture for affective game agents that was tested in a role-playing scenario in terms of content output against traditional LLM architectures (Experiment 2). Finally, the Chain-of-Emotion architecture was implemented into a custom made conversational game to evaluate against No-Memory and Memory architectures in a user evaluation study (Experiment 3). It was found that the Chain-of-Emotion architecture implementing appraisal prompting led to qualitatively different content generations quantified via the LIWC that outperformed the other conditions on multiple user experience items relating to agent believability and observed emotional intelligence of the agent.

Overall this study provides early evidence for the potential of language model agents to understand and simulate emotions in a game context.

## 7.1 Emotional intelligence in language model agents

As more and more evidence arises for the potential of language models to simulate cognitive processes of humans [11], we investigated how this could translate to more affect-focused tasks, specifically emotional intelligence tasks. OpenAI's gpt-3.5 performed well overall in situational emotional labelling, providing some evidence for the utility of such models to identify the most likely emotional reaction for a range of situations. These findings therefore add to the body of evidence indicating that language models could be useful to better understand cognitive processes [66]. Importantly, our findings do not only show that LLMs can solve emotion labelling tasks much better than chance level, but also that the performance is dependent on the underlying prompting strategy. Adapted from successful chain-of-thought prompting strategies [46], we compared prompts without context (No-Memory) to prompts with previously answered questions included (Memory) and to prompts that first ask the model to appraise the situation and then answer the STEU item (appraisal prompting). This third strategy was built upon findings of modern psychological research that show that cognitive appraisal processes are important factors when it comes to human emotion elicitation and understanding [6, 15]. In a recent study by Li et al. [67], emotionally charged prompts have been shown to improve task performance for various language models. Similarly, appraisal-prompting led to better performance in the emotion labelling task compared to the other two conditions. This finding can be considered from two perspectives: first, it shows that commonly observed psychological processes might be represented in language and therefore in large language models, providing more evidence for the utility of such models to simulate human responses [11]. Second, techniques built upon such observed psychological processes can be used to improve language model performance for specific tasks and might therefore be considered when it comes to building architectures for language model agents. Especially this second point could be of relevance when considering how language model implementations could be integrated in the future to solve problem-specific tasks. Since performance can be increased through prompting strategies facilitating few-shot learning [36, 46] and language models demonstrate representations of a range of psychological constructs [13], building prompts on these cognitive processes is likely to yield benefits for various tasks.

From a psychological perspective, appraisal has long been acknowledged to be a central part of emotion formation, involving both conscious and unconscious cognitive processes [6, 15]. In its basic definition, appraisal relates to an individual's relationship to a given event in terms of different variables (such as personal significance, goal congruence, etc. [33]). It is not yet clear what specific variables are of importance and how the process of appraisal interacts with other emotion components on a detailed level [6]. That is to say, appraisal cannot yet be universally modelled and therefore implemented within a computational system. We may however assume that information that makes the appraisal process observable and usable is represented in language and therefore also in LLMs [68]. It can therefore be argued that language models could solve some of the practicality problems present in the discipline of affective computing [1]. If LLMs have the ability to solve EI tasks through mechanisms mirroring appraisal, we can make use of these models to potentially build affective agents [2] without the need to fully solve the remaining theoretical problems in the field of psychology [6]. The use of language models could therefore be considered a more practical solution to producing useful agents, even if open questions regarding the understanding of human emotion remain.

## 7.2 User Interaction with Chain-of-Emotion language model agents

Implementing appraisal prompting into a Chain-of-Emotion system (see Fig 1 for a schematic representation), it was possible to test output contents as measured with the LIWC against

other LLM architecture implementations (see Fig 4 for details on each condition). For the purposes of this study, the implementation was kept as simple as possible and only included a text storage (memory system) and an appraisal-prompted emotion generation (Appraisal System) before character dialog was generated. Within a custom-made role-playing scenario where the agent was used to play out a breakup scenario with their long-term romantic partner, the Chain-of-Emotion architecture demonstrated a higher Authenticity score when prompted with controlled prompts that were kept fixed between all conditions. When tested with players in a custom video game, the Chain-of-Emotion architecture led to a significantly different Tone score of the language, potentially signaling the inclusion of more complex emotional responses as observed in the controlled environment. It is important to note that authenticity was only increased with controlled prompts and tone was only different with non-controlled player generated prompts, meaning that the differences in text-generated content was highly influenced by the in-game context. The texts generated for the fixed prompts (see S1 Table) yielded potentially more complex emotional responses (for example a mix of melancholy and nostalgia) in the Chain-of-Emotion condition compared to the other conditions.

This pattern was also observable within the user ratings. The Chain-of-Emotion agent was rated significantly more natural and responsive than the agent in other conditions, and additionally more sensitive to emotions of others. Other items relating to believability and observed emotional intelligence also showed trends of better performances for the Chain-of-Emotion condition. Building such an architecture has therefore quantifiable benefits when it comes to the user experience of artificial agents, which is one of the most important evaluation criteria, especially in the domain of video games [69]. Importantly, no differences in personality aspect ratings (on the classic domains of competency, warmth, capability, and friendliness) were observed. This could be seen as evidence that all implemented language model agents followed the task of role-playing the given character with the provided personality. But the Chain-of-Emotion architecture outperformed the other architectures in terms of observed emotional intelligence items (quantified via the STEU score) and believability (quantified via user rating). The proposed architecture therefore yielded convincing results on multiple evaluation criteria (qualitative characteristics of content, user-rated believability, user-rated emotional intelligence, in addition to the previously tested emotion understanding) and can therefore be seen as a step towards well-functioning affective language model game agents that could solve some of the problems present in the field [2]. Most importantly, because language model agents have the abilities to simulate human-like cognitive tasks [11], a successful game agent architecture does not need to solve fundamental problems in theoretical psychology before creating computational implementations as previously considered [1, 20, 21]. Rather, a language model agent architecture needs to make use of the characteristics of LLMs and implement systems solving more practical concerns, such as memory tasks (both storing and retrieval [41]), or performance-enhancing tasks, such as the proposed appraisal-prompting step. Future research can expand these efforts and test more complex systems and varying game mechanics and user interfaces.

### 7.3 Limitations

Language models do not simulate the behavior of a human, but provide probable language outputs that in some form represent human behavior. That is to say, models are bound to their statistical capabilities even in a theoretical, infinitely trained model [70]. This means that while there is no doubt of the potential of language models to solve some tasks with human-like performance [11], other tasks (e.g., truth-telling [70] or casual reasoning [11]) can pose difficulties. As human affect is a very complex field with various competing theoretical perspectives,

LLMs cannot be seen as accurate simulation machines of affective human processes. Rather, the provided results show that some psychological processes can be simulated through their representations in language that can be replicated through deep learning techniques. Our application is built specifically upon the process of appraisal [15], which is a well-researched component of human affect. There are however many other processes involved, including physiological, behavioral, and feeling components that are points of contention among theorists [7]. As recently discussed by Croissant, Schofield and McCall [22], it is an important task of affective systems to provide robust theoretical frameworks and technical implementations to not make unfounded theoretical assumptions. While the presented methodology can be seen as robust to theoretical uncertainties, it does not account for the full spectrum of human emotional responses (and not even the spectrum of cognitive aspects of emotional responses). While the presented results seem promising, more work needs to be done to further explore affect simulation with LLM agents.

Another limitation stems from the simplicity of our implementation. To test this new kind of architecture for affect simulation, we chose to make all systems as easy as possible. As shown in the study by Park et al. [41], generative video game agents benefit from certain implementations of memory systems that can store and retrieve information with relevancy for the given situation. Since the tested game was rather short and all interactions had relevancy, we did not include a memory-retrieval step, which might be necessary in longer and more complex games. Similarly, the output was limited to the chosen characteristics and context constraints (i.e., the dating scenario) and any additional challenges posed by other contexts/characterisations prevalent in video games (e.g., fantasy settings, historical settings, science fiction settings, etc.) might need further experimentation. LLM architectures are getting more and more involved with techniques such as chaining [71, 72], dynamic context [73], function calling (and therefore more complex decision making), and more. The experiments presented here were designed to present foundational insights about chain of emotion, but fully realized affective agents need to be embedded in more complex systems and further evaluated in their ability to simulate emotional reactions. Furthermore, this study made use of only one LLM (namely OpenAI's gpt-3.5) as we had no access to other models. As described in some early reports (e.g., [74]), newer models such as GPT-4 likely outperform previous models on various criteria, which likely influences the impact of strategies such as appraisal prompting and Chain-of-Emotion architectures. However, given the domain-specific aims of game character simulation, it can not be assumed that game companies will want to make use of the most powerful language models in every case. Providing strategies for improving language model capabilities will have value in any case and should inform the process of creating and using appropriate models to solve emotion understanding and simulation tasks in the future.

## 8 Conclusion

This study adds to the body of research showcasing the capabilities of LLMs to solve psychological tasks, namely emotional intelligence items and simulation of believable emotions. The affective information represented through the training data in language models seem to hold the necessary information that makes inferring plausible affective states to others possible, which adds to the results showcasing theory-of-mind abilities of LLMs [12]. The performance is dependent on prompting strategies—utilizing appraisal prompting increases emotional intelligence scores. Based on these results, a Chain-of-Emotion architecture for conversational game scenarios was constructed that implemented an appraisal step to simulate emotional responses and used the resulting information for dialog generation. This new architecture improved the agent's performance in believably simulating complex emotions (in addition to

the previously shown benefits for emotion intelligence). The proposed system can therefore be seen as a first step towards affective game agents using language models. Yet, it is still necessary to test and refine such systems and implement them within more complex LLM agents (that include more complex memory systems with chaining and retrieval techniques, as well as more dynamic in-game behavior). Our system does not result in a complete emotion simulation, but in a virtual agent that is perceived as believable and scores high in emotional intelligence tasks. By using the right architecture, language models are therefore likely to be able to simulate emotion-like behavior in artificial agents. By refining the efforts presented here, we could approach fully realized affective artificial agents that might facilitate engaging, immersive, and enjoyable virtual interactions.

## Supporting information

**S1 Table. Instruction, input, and responses for Experiment 2.**
(ZIP)

## Author Contributions

**Conceptualization:** Maximilian Croissant, Madeleine Frister, Guy Schofield, Cade McCall.

**Data curation:** Maximilian Croissant.

**Formal analysis:** Maximilian Croissant.

**Investigation:** Maximilian Croissant.

**Methodology:** Maximilian Croissant.

**Project administration:** Cade McCall.

**Resources:** Madeleine Frister.

**Software:** Maximilian Croissant, Madeleine Frister.

**Validation:** Maximilian Croissant.

**Visualization:** Maximilian Croissant.

**Writing – original draft:** Maximilian Croissant.

**Writing – review & editing:** Maximilian Croissant, Guy Schofield, Cade McCall.

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
