## [Decision Letter · Decision Letter 0]

8 Nov 2023

PONE-D-23-31194An Appraisal-Based Chain-Of-Emotion Architecture for Affective Language Model Game AgentsPLOS ONE

Dear Dr. Croissant,

Thank you for submitting your manuscript to PLOS ONE. After careful consideration, we feel that it has merit but does not fully meet PLOS ONE’s publication criteria as it currently stands. Therefore, we invite you to submit a revised version of the manuscript that addresses the points raised during the review process.

We look forward to receiving your revised manuscript.

Kind regards,

Michal Ptaszynski, PhD

Academic Editor

PLOS ONE

5. We note that Figure 4 in your submission contain copyrighted images. All PLOS content is published under the Creative Commons Attribution License (CC BY 4.0), which means that the manuscript, images, and Supporting Information files will be freely available online, and any third party is permitted to access, download, copy, distribute, and use these materials in any way, even commercially, with proper attribution. For more information, see our copyright guidelines: http://journals.plos.org/plosone/s/licenses-and-copyright.

a. You may seek permission from the original copyright holder of Figure 4 to publish the content specifically under the CC BY 4.0 license. 

6. We notice that your supplementary information are included in the manuscript file. Please remove them and upload them with the file type 'Supporting Information'. Please ensure that each Supporting Information file has a legend listed in the manuscript after the references list.

Reviewers' comments:

Reviewer's Responses to Questions

**Comments to the Author**

1. Is the manuscript technically sound, and do the data support the conclusions?

Reviewer #1: Yes

Reviewer #2: Yes

Reviewer #3: Yes

Reviewer #4: Yes

2. Has the statistical analysis been performed appropriately and rigorously? 

Reviewer #1: Yes

Reviewer #2: Yes

Reviewer #3: Yes

Reviewer #4: Yes

3. Have the authors made all data underlying the findings in their manuscript fully available?

Reviewer #1: Yes

Reviewer #2: Yes

Reviewer #3: Yes

Reviewer #4: Yes

4. Is the manuscript presented in an intelligible fashion and written in standard English?

Reviewer #1: Yes

Reviewer #2: No

Reviewer #3: Yes

Reviewer #4: Yes

5. Review Comments to the Author

Reviewer #1: This paper is based on An Appraisal-Based Chain-Of-Emotion Architecture for Affective Language Model Game Agents. Thus, this paper is directly related to the theme of this journal. Overall, the paper is organized properly; the concept and future research directions are extensively explained. So, the paper is accepted after following minor changes:

1. Problem of paper and motivation is not clear in introduction

2. Contribution of paper is not clear and not given in bullets

3. Paper contains few grammar mistakes which will be cooperated in final version.

4. Only 67 references are added in paper, but more than 75 references so to attract readers add few latest references related to this paper, which is mentioned below

Laghari, Asif Ali, Hui He, Kamran Ali Memon, Rashid Ali Laghari, Imtiaz Ali Halepoto, and Asiya Khan. "Quality of experience (QoE) in cloud gaming models: A review." multiagent and grid systems 15, no. 3 (2019): 289-304.

Laghari, Asif Ali, Kamran Ali Memon, Muhammad Bux Soomro, Rashid Ali Laghari, and Vishal Kumar. "Quality of experience (QoE) assessment of games on workstations and mobile." Entertainment Computing 34 (2020): 100362.

Madiha, Hina, LiHui Lei, Asif Ali Laghari, and Sajida Karim. "Quality of experience and quality of service of gaming services in fog computing." In Proceedings of the 2020 4th international conference on management engineering, software engineering and service sciences, pp. 225-228. 2020.

Laghari, Asif Ali, Sana Shahid, Rahul Yadav, Shahid Karim, Awais Khan, Hang Li, and Yin Shoulin. "The state of art and review on video streaming." Journal of High Speed Networks Preprint (2023): 1-26.

Laghari, Asif Ali, Xiaobo Zhang, Zaffar Ahmed Shaikh, Asiya Khan, Vania V. Estrela, and Saadat Izadi. "A review on quality of experience (QoE) in cloud computing." Journal of Reliable Intelligent Environments (2023): 1-15.

Reviewer #2: - Introduction needs to be revised, including the problem identification and research gaps.

- Methodology is the not discussed systematically.

- Expand the critical results in the conclusion. Focus on the main developments in the finale. Also, write the main contributions in the conclusion.

- Results are not described properly.

- All figures have low quality, and please improve all of them.

- The article needs to be a review of grammatical errors.

Reviewer #3: This paper provides a study that is of great interest in the gaming field and increasingly in the VR field. The comparison of the three level of experiement give a siginificant understanding of future possibilities. However, there are still some parts that could be improve to make this paper more clear and also readable to a bigger audience.

1) For the few studies, are the three studies done on the same experiement? Or are they studies of three different experiments? The whole experiment process is not very concise and spreaded around the whole paper. I would suggest to put them at the end of chapter 3 before going into details.

2) This paper compares three different types of situation, no memory as basis, and then memory and appraisal. Since the key of this paper is about the appraisal and the memory, a bit more explanation and diagrams to show the difference between the two will be very necessary.

3) The language and words might be also an affecting factors, it might not be part of the key research of this paper, but I believe some explanation or observation should be given to explain how the results are putting this aside at this stage.

4) The design of the UI as well, it is good that the author provide a screenshot but no explanation was done with respect to the interface. There will definitely be some influence in what the participants see on the screen as well. It will be better if the author put that into account for the discussion.

Reviewer #4: The paper explores using large language models (LLMs) to develop believable and interactive artificial agents that simulate human emotions. Based on psychological appraisal research, the study presents a new chain-of-emotion architecture for emotion simulation in video games, which outperforms standard LLM architectures in user experience and content analysis metrics, Below are my comments.

Comment #1:

The paper should clarify the LLMs used in the study and expand on the testing scenarios. A description of the models' training datasets, limitations, and capabilities would greatly help in assessing the validity of the results. Testing the proposed Chain-of-Emotion architecture across varied gaming genres and contexts is recommended to strengthen the argument of its general applicability.

Comment #2:

The claim that the new architecture outperforms "standard LLM architectures" lacks a clear benchmark. The study should specify the other LLM architectures used for comparison, detailing their design and the metrics on which they were evaluated. This would provide the reader with a clearer understanding of the proposed architecture's relative performance.

Comment #3:

While user-rated metrics provide valuable insights into the user experience, the study should aim to balance these with objective measures where possible. The diversity and representativeness of the user group should be disclosed to ensure the reliability of these subjective metrics.

Comment #4:

The paper could further acknowledge the complexity of human emotions and how this complexity poses a challenge for AI simulation. There could be a discussion about how architecture accounts for or falls short in simulating the full spectrum of human emotional responses.

Comment #5:

It is suggested that the study references a broader range of psychological theories to ensure that the architecture isn't overly dependent on a narrow set of assumptions. The implications of basing LLM outcomes on these observations should be thoroughly discussed.

Comment #6:

The study should provide a clear, transparent methodology that allows other researchers to replicate the work. This includes detailed descriptions of the game scenarios used, the nature of the emotional responses evaluated, and the precise nature of the chain-of-emotion mechanism.

Comment #7:

Additionally, how to enhance the game agent architecture to better compete with other LLM frameworks such as langChan, Llama Index, Autogen, etc. that boast advanced modules like memory, chains, agents, callbacks, security, and integration capabilities:

• While the architecture's focus on emotional intelligence within conversational agents is commendable, it could benefit from incorporating more sophisticated memory modules, similar to those seen in competing frameworks. Enhanced memory capabilities would allow for a more nuanced understanding of context over longer interactions, which is crucial for maintaining coherent and emotionally appropriate responses. Consider adopting or developing memory structures that can handle complex conversational threads without losing the emotional thread of the interaction.

• The game agent architecture has potential, but to truly compete, it should look into creating or integrating more complex chain mechanisms. Chains that can manage sequences of interrelated tasks would provide a significant edge, enabling agents to handle multiturn dialogues with more awareness and anticipation of user needs, thus improving the emotional engagement in conversations.

• Your approach to developing emotional intelligence in game agents is innovative; however, it might be beneficial to include more robust agent management and callback functions that other frameworks offer. These would allow for better event-driven interactions, which can result in more dynamic and responsive emotional behaviors in real-time, leading to a more immersive user experience.

• Security is an increasingly vital concern in AI applications. To enhance the competitive edge of your architecture, it is crucial to integrate state-of-the-art security protocols to ensure user data, especially as it pertains to emotional data, is handled with the utmost care. This would not only increase trust in your system but also align with best practices in responsible AI development.

• Integration capabilities are a standout feature in existing LLM frameworks. To bolster your architecture's marketability, provide clear and streamlined processes for integrating with popular LLMs, databases, and external APIs. Ensuring your system can easily fit within different tech stacks will be key to its adoption.

• Lastly, the game agent architecture should refine its context management system. While handling emotional responses is your architecture's unique selling point, the ability to maintain and leverage context effectively over the course of long interactions is what will truly enhance its practicality and appeal. Better context management can lead to more personalized and accurate emotional interactions, which is paramount for user engagement.

By addressing these comments, the study can significantly improve its scientific rigor, relevance, and potential impact in the field of AI-driven emotional simulation within interactive media.

6. PLOS authors have the option to publish the peer review history of their article (what does this mean?). If published, this will include your full peer review and any attached files.

Reviewer #1: No

Reviewer #2: **Yes: **krishna kumar mohbey

Reviewer #3: **Yes: **Sky Lo

Reviewer #4: No

---

## [Author Response · Author response to Decision Letter 0]

21 Dec 2023

Editor Comments

Thank you for the comments and suggestions. We have changed the manuscript accordingly and believe the changes improve the manuscript. Please find attached detailed answers for each comment raised. We fixed formatting issues, styling, inconsistent funding information, and supplementary material placement. 

Regarding code sharing: We’re happy to make the Unity game project publicly available. Nevertheless, we believe our main contribution is a high-level architecture described within the paper that is agnostic to specific code or programming languages. The code used was a specific implementation in a game example, but we wish the architecture to be viewed independent from this specific implementation. 

Regarding Figure 4: This figure represents an in-game screenshot created by the authors. All authors grant permission to use the figures under the CC BY 4.0 license. This is not a copyrighted image of a published game, it is purely a screenshot from the used experiment created by the authors. Please let us know if a content permission form is necessary in this case. 

Sincerely,

Maxi Croissant

Reviewer #1

1. Problem of paper and motivation is not clear in introduction

Thank you for your comments. We’ve made some changed to the introduction to better reflect the motivation and make the whole process clearer.

2. Contribution of paper is not clear and not given in bullets

We hope our changes to the conclusion make our contributions clear and easy to understand. 

3. Paper contains few grammar mistakes which will be cooperated in final version.

Yes – we hope to have addressed these oversights.

4. Only 67 references are added in paper, but more than 75 references so to attract readers add few latest references related to this paper, which is mentioned below

Laghari, Asif Ali, Hui He, Kamran Ali Memon, Rashid Ali Laghari, Imtiaz Ali Halepoto, and Asiya Khan. "Quality of experience (QoE) in cloud gaming models: A review." multiagent and grid systems 15, no. 3 (2019): 289-304.

Laghari, Asif Ali, Kamran Ali Memon, Muhammad Bux Soomro, Rashid Ali Laghari, and Vishal Kumar. "Quality of experience (QoE) assessment of games on workstations and mobile." Entertainment Computing 34 (2020): 100362.

Madiha, Hina, LiHui Lei, Asif Ali Laghari, and Sajida Karim. "Quality of experience and quality of service of gaming services in fog computing." In Proceedings of the 2020 4th international conference on management engineering, software engineering and service sciences, pp. 225-228. 2020.

Laghari, Asif Ali, Sana Shahid, Rahul Yadav, Shahid Karim, Awais Khan, Hang Li, and Yin Shoulin. "The state of art and review on video streaming." Journal of High Speed Networks Preprint (2023): 1-26.

Laghari, Asif Ali, Xiaobo Zhang, Zaffar Ahmed Shaikh, Asiya Khan, Vania V. Estrela, and Saadat Izadi. "A review on quality of experience (QoE) in cloud computing." Journal of Reliable Intelligent Environments (2023): 1-15.

We’ve added more relevant references as other reviewers also wished more context for certain points. We hope this addresses all concerns of missing context.

Reviewer #2

- Introduction needs to be revised, including the problem identification and research gaps.

Thank you for your comments. We’ve revised the introduction to better state our motivation and potential contribution to the field.

- Methodology is the not discussed systematically.

We hope our changes to the methods sections in all experiments enhance clarity and replicability (specifically our new description of individual conditions with a new figure 4).

- Expand the critical results in the conclusion. Focus on the main developments in the finale. Also, write the main contributions in the conclusion.

Yes – we hope our changes to the conclusion address these concerns.

- Results are not described properly.

We’ve changed how we present results to clearly provide differences in all conditions for all experiments.

- All figures have low quality, and please improve all of them.

We’ve made some changes to the figures. Regarding quality, these were created using the PLOS ONE tooling and we hope these are in accordance with the journal styling requirements.

- The article needs to be a review of grammatical errors.

Yes – we have addressed all errors in the revision.

Reviewer #3

1) For the few studies, are the three studies done on the same experiement? Or are they studies of three different experiments? The whole experiment process is not very concise and spreaded around the whole paper. I would suggest to put them at the end of chapter 3 before going into details.

Thank you for your comments. The three studies are three separate experiments. We’ve changed the titles of the sections in the paper to reflect this and added an explanation at the end of section 3 to explain the overall process better and give a clearer structure to the process. 

2) This paper compares three different types of situation, no memory as basis, and then memory and appraisal. Since the key of this paper is about the appraisal and the memory, a bit more explanation and diagrams to show the difference between the two will be very necessary.

We’ve added an additional figure to demonstrate how the three architectures differ from each other (Fig 4) and added more explanations for the three approaches in section 5.

3) The language and words might be also an affecting factors, it might not be part of the key research of this paper, but I believe some explanation or observation should be given to explain how the results are putting this aside at this stage.

This is true. Since LLMs work exclusively with language, this is a big factor to consider. This is why Experiment 1 and Experiment 2 are highly controlled for language variance – all three architectures are tested with exactly the same prompts. Experiment 3 adds variation due to different users giving different input, which is addressed through a large enough sample to ensure high power for a medium-sized effect. We test both the controlled and the more open scenario of a real-life implementation to assess the potential for affective LLM agents. 

We hope to better explain this in the changes made to the methods sections, specifically Experiment 2.

4) The design of the UI as well, it is good that the author provide a screenshot but no explanation was done with respect to the interface. There will definitely be some influence in what the participants see on the screen as well. It will be better if the author put that into account for the discussion.

Thank you – again this is true. We’ve expanded the methods section of Experiment 3 to better describe the user interface (Section 6.1.1). The screenshots do show the entire UI – there is only a text field and a button. We have tried to make this clearer with those changes. How much the UI influences the effects has not been tested. This is a potential topic for future research, which we now mention in the Discussion (Section 7.2). 

Reviewer #4

Comment #1:

The paper should clarify the LLMs used in the study and expand on the testing scenarios. A description of the models' training datasets, limitations, and capabilities would greatly help in assessing the validity of the results. Testing the proposed Chain-of-Emotion architecture across varied gaming genres and contexts is recommended to strengthen the argument of its general applicability.

Thank you for your comments. We tried to address the clarification issues through an additional Figure (Fig 4) and more explanations regarding the differences between architectures (see Section 5.1). We’ve also included more details to the limitations and parameters of the LLM used. We agree that the architecture needs to be tested in further scenarios – here we present 3 distinct experiments building on each other to validate the architecture as a way to create affective agents. We feel like this reaches the limit of what we can do in one contained study, but we expanded our discussion (see 7.2 and 7.3) for the need of further gaming-related applications to better understand how applicable the architecture could be. 

Comment #2:

The claim that the new architecture outperforms "standard LLM architectures" lacks a clear benchmark. The study should specify the other LLM architectures used for comparison, detailing their design and the metrics on which they were evaluated. This would provide the reader with a clearer understanding of the proposed architecture's relative performance.

This statement refers specifically to the new architecture outperforming both control groups, and it has been slights adapted to reflect this. It was our aim to clearly describe methodology and benchmark results for each experiment. We’ve made some changes to the presentations of our methods and results to give more details of (1) how each architecture functions, and (2) how well it performs in each task, including providing statistical comparisons wherever possible.

Comment #3:

While user-rated metrics provide valuable insights into the user experience, the study should aim to balance these with objective measures where possible. The diversity and representativeness of the user group should be disclosed to ensure the reliability of these subjective metrics.

We agree. This is why we included 3 distinct experiments, all focusing on different aspects of evaluating the proposed architecture. Experiment 1 uses purely objective measures through a validated emotional intelligence instrument, while Experiment 2 uses a mixed-method approach to content analysis. Experiment 3 uses subjective measures to draw conclusions of player experience (PX), which is arguably the most important aspect of game design. We feel like together, these experiments make an effort to balance various objective and subjective outcomes to give a broad picture of the area. We describe demographic data of the participants in section 6.1.5. The age ranges from 19 to 47 years and 33% of participants identified as female. We hope this addresses diversity concerns, although we were not able to fully control for this, as we were limited to the University subject pool. 

Comment #4:

The paper could further acknowledge the complexity of human emotions and how this complexity poses a challenge for AI simulation. There could be a discussion about how architecture accounts for or falls short in simulating the full spectrum of human emotional responses.

This is a very good point and quite crucial for this technology. Human emotion is extremely complex and whatever is simulated within our experiments should not be confused with real affective processes. We added more explanation and further references in our discussion of the limitations to address this further. 

Comment #5:

It is suggested that the study references a broader range of psychological theories to ensure that the architecture isn't overly dependent on a narrow set of assumptions. The implications of basing LLM outcomes on these observations should be thoroughly discussed.

We hope the changes mentioned in comment #4 also address these concerns. It is very important to us to not simplify emotional processes and to not misrepresent the various theoretical perspectives of emotion research. 

Comment #6:

The study should provide a clear, transparent methodology that allows other researchers to replicate the work. This includes detailed descriptions of the game scenarios used, the nature of the emotional responses evaluated, and the precise nature of the chain-of-emotion mechanism.

We agree and we have edited the Method sections to provide more detail. 

Comment #7:

Additionally, how to enhance the game agent architecture to better compete with other LLM frameworks such as langChan, Llama Index, Autogen, etc. that boast advanced modules like memory, chains, agents, callbacks, security, and integration capabilities:

Yes – this is very important. Given that the field is rapidly developing, our main aim was to present the simplest version of the architecture possible (i.e. short enough scenarios to fully be stored in context, no need for chains, and no need for retrieval systems). However, we fully believe that this should be tested within more complex LLM frameworks. We hope our changes made in the discussion will lead to further research approaching these issues.

• While the architecture's focus on emotional intelligence within conversational agents is commendable, it could benefit from incorporating more sophisticated memory modules, similar to those seen in competing frameworks. Enhanced memory capabilities would allow for a more nuanced understanding of context over longer interactions, which is crucial for maintaining coherent and emotionally appropriate responses. Consider adopting or developing memory structures that can handle complex conversational threads without losing the emotional thread of the interaction.

Yes – specifically for memory, there is a lot of interesting work being done involving custom retrieval modules and dynamic contexts. We hope to further test this in future studies. 

• The game agent architecture has potential, but to truly compete, it should look into creating or integrating more complex chain mechanisms. Chains that can manage sequences of interrelated tasks would provide a significant edge, enabling agents to handle multiturn dialogues with more awareness and anticipation of user needs, thus improving the emotional engagement in conversations.

We agree with this point as well. In it’s core, the Chain-Of-Emotion architecture does implement an appraisal-based chaining mechanism that was tested by itself, but the next logical step would be to integrate this with more complex chains – and even agent tasks.

• Your approach to developing emotional intelligence in game agents is innovative; however, it might be beneficial to include more robust agent management and callback functions that other frameworks offer. These would allow for better event-driven interactions, which can result in more dynamic and responsive emotional behaviors in real-time, leading to a more immersive user experience.

• Security is an increasingly vital concern in AI applications. To enhance the competitive edge of your architecture, it is crucial to integrate state-of-the-art security protocols to ensure user data, especially as it pertains to emotional data, is handled with the utmost care. This would not only increase trust in your system but also align with best practices in responsible AI development.

These two steps specifically are very high-level and would provide valuable insights for real-world applications of our proposed architectures. We hope to integrate the approach with concern to such systems in future research. 

• Integration capabilities are a standout feature in existing LLM frameworks. To bolster your architecture's marketability, provide clear and streamlined processes for integrating with popular LLMs, databases, and external APIs. Ensuring your system can easily fit within different tech stacks will be key to its adoption.

• Lastly, the game agent architecture should refine its context management system. While handling emotional responses is your architecture's unique selling point, the ability to maintain and leverage context effectively over the course of long interactions is what will truly enhance its practicality and appeal. Better context management can lead to more personalized and accurate emotional interactions, which is paramount for user engagement.

These points as well as very relevant. Since receiving this review, OpenAI has already introduced the new Assistants API and common frameworks and stacks are currently constantly adopted to address the rapid changes in the field. Our system might not yet integrate with many of these tools, because of our (for now) only fundamental research. This has the benefit of being agnostic to these rapid changes, which is important for introducing such techniques, but we hope to integrate with modern tech stacks in the future.

Overall we agree on all these points and thank you for this thorough write up. All these mechanisms would have introduced potential sources of variability at th

---

## [Decision Letter · Decision Letter 1]

11 Mar 2024

An Appraisal-Based Chain-Of-Emotion Architecture for Affective Language Model Game Agents

PONE-D-23-31194R1

Dear Dr. Croissant,

We’re pleased to inform you that your manuscript has been judged scientifically suitable for publication and will be formally accepted for publication once it meets all outstanding technical requirements.

Kind regards,

Michal Ptaszynski, PhD

Academic Editor

PLOS ONE

Additional Editor Comments (optional):

Reviewers' comments:

Reviewer's Responses to Questions

**Comments to the Author**

1. If the authors have adequately addressed your comments raised in a previous round of review and you feel that this manuscript is now acceptable for publication, you may indicate that here to bypass the “Comments to the Author” section, enter your conflict of interest statement in the “Confidential to Editor” section, and submit your "Accept" recommendation.

Reviewer #1: All comments have been addressed

Reviewer #2: All comments have been addressed

Reviewer #4: All comments have been addressed

2. Is the manuscript technically sound, and do the data support the conclusions?

Reviewer #1: Yes

Reviewer #2: Yes

Reviewer #4: Yes

3. Has the statistical analysis been performed appropriately and rigorously? 

Reviewer #1: Yes

Reviewer #2: N/A

Reviewer #4: Yes

4. Have the authors made all data underlying the findings in their manuscript fully available?

Reviewer #1: Yes

Reviewer #2: Yes

Reviewer #4: Yes

5. Is the manuscript presented in an intelligible fashion and written in standard English?

Reviewer #1: Yes

Reviewer #2: Yes

Reviewer #4: Yes

6. Review Comments to the Author

Reviewer #1: authors done good work, quality of paper is improved and revised paper according to suggestions so I recommend this paper as accept

Reviewer #2: (No Response)

Reviewer #4: The author has meticulously addressed every one of my comments and concerns, demonstrating a thorough and thoughtful consideration of the feedback provided.

7. PLOS authors have the option to publish the peer review history of their article (what does this mean?). If published, this will include your full peer review and any attached files.

Reviewer #1: No

Reviewer #2: No

Reviewer #4: No

---

## [Editor Report · Acceptance letter]

29 Apr 2024

PONE-D-23-31194R1 

PLOS ONE

Dear Dr. Croissant, 

I'm pleased to inform you that your manuscript has been deemed suitable for publication in PLOS ONE. Congratulations! Your manuscript is now being handed over to our production team.

Kind regards, 

on behalf of

Dr. Michal Ptaszynski 

Academic Editor

PLOS ONE